# Correcting Nuisance Variation using Wasserstein Distance

## Abstract

Profiling cellular phenotypes from microscopic imaging can provide meaningful biological information resulting from various factors affecting the cells. One motivating application is drug development: morphological cell features can be captured from images, from which similarities between different drugs applied at different dosages can be quantified. The general approach is to find a function mapping the images to an embedding space of manageable dimensionality whose geometry captures relevant features of the input images. An important known issue for such methods is separating relevant biological signal from nuisance variation. For example, the embedding vectors tend to be more correlated for cells that were cultured and imaged during the same week than for cells from a different week, despite having identical drug compounds applied in both cases. In this case, the particular batch a set of experiments were conducted in constitutes the domain of the data; an ideal set of image embeddings should contain only the relevant biological information (e.g. drug effects). We develop a general framework for adjusting the image embeddings in order to 'forget' domain-specific information while preserving relevant biological information. To do this, we minimize a loss function based on distances between marginal distributions (such as the Wasserstein distance) of embeddings across domains for each replicated treatment. For the dataset presented, the replicated treatment is the negative control. We find that for our transformed embeddings (1) the underlying geometric structure is not only preserved but the embeddings also carry improved biological signal (2) less domain-specific information is present.

## 1 Introduction

In the framework where our approach is applicable, there are some inputs (e.g. images) and a map $\mathcal{F}$ sending the inputs to vectors in a low-dimensional space which summarizes information about the inputs. $\mathcal{F}$ could either be engineered using specific image features, or learned (e.g. using deep neural networks). We will call these vectors 'embeddings' and the space to which they belong the 'embedding space'. Each input may also have corresponding semantic labels and domains, and for inputs with each label and domain pair, $\mathcal{F}$ produces some distribution of embeddings. Semantically meaningful similarities between pairs of inputs can then be assessed by the distance between their corresponding embeddings, using some chosen distance metric. Ideally, the embedding distribution of a group of inputs depends only on their label, but often the domain can influence the embedding distribution as well. We wish to find an additional map to adjust the embeddings produced by $\mathcal{F}$ so that the distribution of adjusted embeddings for a given label is independent of the domain, while still preserving semantically meaningful distances between distributions of inputs with different labels.

The map $\mathcal{F}$ can be used for phenotypic profiling of cells. In this application, images of biological cells perturbed by one of several possible biological stimuli (e.g. various drug compounds at different doses, some of which may have unknown effects) are mapped to embeddings, which are used to reveal similarities among the applied perturbations.

There are a number of ways to extract embeddings from images of cells. One class of methods such as that used by Ljosa et al. (2013) relies on extracting specifically engineered features. In the recent work by Ando et al. (2017), a Deep Metric Network pre-trained on consumer photographic images (not microscope images of cells) described in Wang et al. (2014) was used to generate embedding

vectors from cellular images, and it was shown that these clustered drug compounds by their mechanisms of action (MOA) more effectively. See Figure 1 for example images of the different MOAs.

Currently one of the most important issues with using image embeddings to discriminate the effects of each treatment (i.e. a particular dose of a drug, the 'label' in the general problem described above) on morphological cell features is nuisance factors related to slight uncontrollable variations in each biological experiment. Many cell imaging experiments are organized into a number of batches of experiments occurring over time, each of which contains a number of sample plates (typically 3-6), each of which contains individual wells in which thousands of cells are grown and treatments are applied (typically around 96 wells per plate). For this application, the 'domain' is an instance of one of these hierarchical levels, and embeddings for cells with a given treatment tend to be closer to each other within the same domain than from a different one. For example, the experimentalist may apply slightly different concentrations or amounts of a drug compound in two wells in which the same treatment was anticipated. Another example is the location of a particular well within a plate or the order of the plate within a batch, which may influence the rate of evaporation, and hence, the appearance of the cells. Finally, 'batch' effects may result from differences in experiment conditions (temperature, humidity) from week to week; they are various instances of this hierarchical level that we will consider as 'domains' in this work.

Our approach addresses the issue of nuisance variation in embeddings by transforming the embedding space in a possibly domain-specific way in order to minimize the variation across domains for a given treatment. We remark that our main goal is to introduce a general flexible framework to address this problem. In this framework, we use a metric function measuring the distances among pairs of probability distributions to construct an optimization problem whose solution yields appropriate transformations on each domain. In our present implementation, the Wasserstein distance is used as a demonstration of a specific choice of the metric that can yield substantial improvements. The Wasserstein distance makes few assumptions about the probability distributions of the embedding vectors.

Our approach is fundamentally different than those which explicitly identify a fixed 'target' and 'source' distributions. Instead, we incorporate information from all domains on an equal footing, transforming all the embeddings. This potentially allows our method to incorporate several replicates of a treatment across different domains to learn the transformations, and not only the controls. We highlight that other distances may be used in our framework, such as the Cramer distance. This may be preferable since the Cramer distance has unbiased sample gradients (Bellemare et al., 2017). This could reduce the number of steps required to adjust the Wasserstein distance approximation for each step of training the embedding transformation. Additionally we propose several other extensions and variations in Section 4.1.

## 2 METHOD

### 2.1 PROBLEM DESCRIPTION

Denote the embedding vectors $x_{t,d,p}$ for $t \in T$, $d \in D$, and $p \in I_{t,d}$, where $T$ and $D$ are the treatment and domain labels respectively, and $I_{t,d}$ is the set of indices for embeddings belonging to treatment $t$ and domain $d$. Suppose that $x_{t,d,p}$ were sampled from a probability distribution $\nu_{t,d}$. our goal is to 'forget' the nuisance variation in the embeddings, which we formalize in the following way. We wish to find maps $A_d$ transforming the embedding vectors such that the transformed marginals $\tilde{\nu}_{t,d}$ have the property that for each $t \in T$ and $d_i, d_j \in D$, $\tilde{\nu}_{t,d_i} \approx \tilde{\nu}_{t,d_j}$ (for some suitable metric between distributions). Intuitively, the transformations $A_d$ can be thought of as correcting a domain-specific perturbation. We do not have 'source' and 'target' distributions, and instead perturb all the embedding distributions simultaneously. The transformations $A_d$ should be small to avoid distorting the underlying geometry of the embedding space, since we do not expect nuisance variation to be very large.

## 2.2 GENERAL APPROACH

The 1-Wasserstein distance (hereafter will be simply referred to as the Wasserstein distance) between two probability distributions $\nu_r$ and $\nu_g$ on a compact metric space $\chi$ with metric $\delta$ is given by

$$W(\nu_r, \nu_g) = \inf_{\gamma \in \Pi(\nu_r, \nu_g)} E_{(x,y) \sim \gamma} \delta(x, y). \tag{1}$$

Here $\Pi(\nu_r, \nu_g)$ is the set of all joint distributions $\gamma(x, y)$ whose marginals are $\nu_r$ and $\nu_g$. This can be intuitively interpreted as the minimal cost of a transportation plan between the probability masses of $\nu_r$ and $\nu_g$. In our application, the metric space was $\mathbb{R}^n$ and $\delta$ was the Euclidean metric. If the Wasserstein distance between two distributions is zero, then it becomes impossible to discern the origin of a sample from one of these two distributions. In addition, the Wasserstein distance (as well as other related metrics for probability distributions) are more appropriate to use than classifiers. This is because classifiers are more sensitive to the distinguishability between probability distributions than other potentially meaningful features. For instance, two otherwise identical Gaussian distributions displaced from one another would have Wasserstein distance equal to the displacement between them. On the contrary, a classifier would yield a function that has vanishing gradients for sufficiently large displacement.

Given two or more probability distributions, their mean can be defined under the Wasserstein distance, known as the 'Wasserstein barycenter'. Explicitly, the Wasserstein barycenter of $N$ distributions $\nu_1, ..., \nu_N$ is defined as the distribution $\mu$ that minimizes

$$\frac{1}{N} \sum_{i=1}^{N} W(\mu, \nu_i). \tag{2}$$

The Wasserstein barycenter and its computation have been studied in many contexts, such as optimal transport theory (Cuturi & Doucet, 2014; Anderes et al., 2016). In Tabak & Trigila (2018), the Wasserstein barycenter has been suggested as a method to remove nuisance variation in high-throughput biological experiments. Two key ingredients of the Wasserstein barycenter are that (i) the nuisance variation is removed in the sense that a number of distinct distributions are transformed into a common distribution, and hence become indistinguishable; and (ii) the distributions are minimally perturbed by the transformtions.

Our method is based on these two requirements, where a separate map is associated with each domain. For each treatment, the average Wasserstein distance among all pairs of transformed distributions across domains is included in the loss function. Specifically, the average Wasserstein distance is formulated as

$$\frac{2}{N(N-1)} \sum_{i,j=1, i<j}^{N} W(A_{d_i}(\nu_i), A_{d_j}(\nu_j)), \tag{3}$$

where the coefficient is the normalizing constant. When multiple treatments are considered, the same number of average Wasserstein distances corresponding to the treatments are included in the loss function. Thus, (i) is achieved by minimizing a loss function containing pairwise Wasserstein distances. Compared with the ResNet used in Shaham et al. (2017), we achieve (ii) by early stopping or adding a regularization term to the loss function. In Section 4.1, we will present another possible formulation that aligns more closely with the idea of the Wasserstein barycenter.

One distinct advantage of the Wasserstein distance is that this metric avoids problematic vanishing gradients during training, which are known to occur for metrics based on the KL-divergence, such as the cross entropy (Arjovsky et al., 2017). This is important from a practical point of view because vanishing gradients may halt the solving of the resulting minimax problem in our method.

The Wasserstein distance does not have a closed form except for a few special cases, and must be approximated in some way. The Wasserstein distance is closely related to the maximum mean discrepancy (MMD) approximated in Shaham et al. (2017) using an empirical estimator based on the kernel method. This method requires selecting a kernel and relevant parameters. In our application, we do not have a fixed 'target' distribution, so the kernel parameters would have to be updated during training. We choose instead to use a method based on the ideas in Arjovsky et al. (2017) and Gulrajani et al. (2017) to train a neural network to estimate the Wasserstein distance. A similar

approach has been proposed in Shen et al. (2017) for domain adaptation. To do this, first apply the Kantorovich-Rubinstein duality:

$$W(\nu_r, \nu_g) = \sup_{\|f\|_L \leq 1} E_{x \sim \nu_r}[f(x)] - E_{x \sim \nu_g}[f(x)]. \tag{4}$$

Here, $\nu_r$ and $\nu_g$ are two probability distributions. The function $f$ is in the space of Lipschitz functions with Lipschitz constant at most 1. To estimate the Wasserstein distance, a function $f$ can be optimized while keeping the norm of its gradient to be less than one. We will call $f$ the 'Wasserstein function' throughout this manuscript.

## 2.3 NETWORK ARCHITECTURE

### 2.3.1 DOMAIN-SPECIFIC TRANSFORMATION

As a preprocessing step, we transform the embeddings for the dataset of interest such that the embeddings for the negative controls have mean zero and an identity covariance matrix (see Section 3.1 for details). We observe that the embeddings for wells corresponding to different dosages of each compound are all shifted away from the origin in roughly the same direction by an amount that generally increases with dosage. The variances of embeddings along the largest principal axes also increase in a manner consistent with the drugs inducing an affine transformation of the embeddings.

Given these observations, we choose to model the impact of nuisance variation by affine transformations, the intuition being that we can treat nuisance variations as small, random, drug-like perturbations resulting from unobserved covariates. It is worth mentioning that we do not expect this assumption to hold generally.

In the current implementation, the domain-specific transformations $A_d$ map input embeddings to transformed embeddings of the same dimension. Each $A_d$ is formulated as an affine transformation $A_d(x) = M_d x + b_d$.

### 2.3.2 LOSS FUNCTION

Collectively denote the parameters for the transformations $A_d$ by $\theta_T$. If a particular treatment $t$ is replicated across two or more domains $d_1, d_2, ..., d_k$, the Wasserstein distances among the transformed distributions are estimated for all same-treatment domain pairs. Notice the parameters for estimating the Wasserstein distance for each $t$ and pair $d_i, d_j$ are different. Collectively denote all Wasserstein estimation parameters by $\theta_W$. We consider the loss function

$$L(\theta_T, \theta_W) = \frac{1}{|T|} \sum_{t \in T} \frac{2}{M_t(M_t - 1)} \sum_{d_i, d_j \in D, i \neq j} \left[ W_{t,d_i,d_j}(\theta_T, \theta_W) - g_{t,d_i,d_j}(\theta_T, \theta_W) \right] + R(\theta_T). \tag{5}$$

In (eq. 5), $W_{t,d_i,d_j}(\theta_T, \theta_W) - g_{t,d_i,d_j}(\theta_T, \theta_W)$ is a penalized approximation to the Wasserstein distance between domains $d_i$ and $d_j$, the function $R(\theta_T)$ is a regularization term for the learned transformation whose purpose is to preserve the geometry of the original embeddings, $M_t$ denotes the number of domains in which treatment $t$ appears, and $|\cdot|$ represents the cardinality of a set.

In this paper, we explore either (i) neglecting $R$ entirely and relying on early stopping instead, and (ii) specifying $R$ as described below and in (eq. 6). Using one of these methods is necessary since otherwise optimizing $L$ may result in embeddings which contain no treatment information (for example, if all embeddings are transformed to a single point).

There may be many possible forms for $R$, and these could involve multiple parameter choices for different components of the transformation that $\theta_T$ determines. In our case, $\theta_T$ parameterizes an affine transformation, and hence we choose

$$R(\theta_T) = \frac{1}{|D|} \sum_d \left( \frac{1}{q} \lambda_M \|M_d\|_F^2 + \lambda_b \|b_d\|_2^2 \right), \tag{6}$$

where $\|\cdot\|_F$ denotes the Frobenius norm, $\|\cdot\|_2$ denotes the $\ell^2$ norm, and $q$ denotes the embedding dimensionality. Moreover, there are two regularization weights $\lambda_M$ and $\lambda_b$.

In (eq. 5), $W_{t,d_i,d_j}$ is used to approximate the Wasserstein distance between the transformed embeddings of domains $d_i$ and $d_j$ for treatment $t$ upon optimization over $\theta_W$. The Wasserstein distance is given by

$$W_{t,d_i,d_j}(\theta_T, \theta_W) = \frac{1}{N} \sum_{p \in I_{t,d_i}} f_{t,d_i,d_j}(A_{d_i}(x_{t,d_i,p}; \theta_T); \theta_W) \tag{7}$$
$$- \frac{1}{N} \sum_{q \in I_{t,d_j}} f_{t,d_i,d_j}(A_{d_j}(x_{t,d_j,q}; \theta_T); \theta_W).$$

Each Wasserstein function $f_{t,d_i,d_j}$ in (eq. 7) depends on the parameters $\theta_W$, while each transformation $A_d$ depends on the parameters $\theta_T$. For simplicity, we assume that $N = |I_{t,d_i}| = |I_{t,d_j}|$, where $|\cdot|$ represents the cardinality of a set. This is a reasonable assumption because in practice, the sets $I_{t,d}$ are chosen as minibatches in stochastic gradient descent. Each of the terms $g_{t,d_i,d_j}$ is a gradient penalty defined in (eq. 8-10).

Each Wasserstein function should be Lipschitz with Lipschitz constant 1. For differentiable functions, this is equivalent to the norm being bounded by 1 everywhere. We use an approach based on Gulrajani et al. (2017) to impose a soft constraint on the norm of the gradient. In this approach, the hard constraint is replaced by a penalty, which is a function of the gradient of the Wasserstein function evaluated at some set of points. The penalty term is weighted by an additional parameter $\gamma$. We find that the value of $\gamma = 10$ used in Gulrajani et al. (2017) works well in our application, and fix it throughout. We remark this is an appropriate choice since it is large enough so that the approximation error in the Wasserstein function is small, while not causing numerical difficulties in the optimization routine. Since it is impossible to check the gradient everywhere, we use the same strategy as Gulrajani et al. (2017): choose the intermediate points $\epsilon A_{d_i}(x_{t,d_i,p_k}; \theta_T) + (1-\epsilon) A_{d_j}(x_{t,d_j,q_k}; \theta_T)$ randomly, where $\epsilon \in U[0,1]$ and $p_k$ and $q_k$ denote the $k_{th}$ element of $I_{t,d_i}$ and $I_{t,d_j}$, respectively. Denote the set of intermediate points by $J_{t,d_i,d_j}$. Intuitively, the reason for sampling along these paths is that the Wasserstein function $f$ whose gradient must be constrained has the interpretation of characterizing the optimal transport between the two probability distributions, and therefore it is most important for the gradient constraint to hold in the intermediate region between the distributions. This is motivated more formally by Proposition 1 in Gulrajani et al. (2017), which shows that an optimal transport plan occurs along straight lines with gradient norm 1 connecting coupled points between the probability distributions. Unlike Gulrajani et al. (2017), we impose the gradient penalty only if the gradient norm is greater than 1. Doing so works better in practice for our application.

Explicitly, we define each gradient penalty $g_{t,d_i,d_j}$ as

$$g_{t,d_i,d_j}(\theta_T, \theta_W) = \frac{1}{N} \sum_{z \in J_{t,d_i,d_j}} H_{t,d_i,d_j}(z; \theta_W), \tag{8}$$

where

$$H_{t,d_i,d_j}(z; \theta_W) = \begin{cases} \gamma \left( G_{t,d_i,d_j}(z; \theta_W) - 1 \right)^2 & \text{if } G_{t,d_i,d_j}(z; \theta_W) > 1, \\ 0 & \text{otherwise.} \end{cases} \tag{9}$$

$$G_{t,d_i,d_j}(z; \theta_W) = \|\nabla_{\theta_W} f_{t,d_i,d_j}(z; \theta_W)\|_2. \tag{10}$$

To approximate the Wasserstein distance we must maximize over $\theta_W$. Thus, our objective is to find

$$\hat{\theta}_T, \hat{\theta}_W = \text{argmin}_{\theta_T} \text{argmax}_{\theta_W} L(\theta_T, \theta_W). \tag{11}$$

We use the approach of Ganin & Lempitsky (2015) to transform our minimax problem to a minimization problem by adding a 'gradient reversal' between the transformed embeddings and the approximated Wasserstein distances. The gradient reversal is the identity in the forward direction, but negates the gradients used for backpropagation.

## 3  EMPIRICAL RESULTS

The embeddings under consideration are generated using the method described in Ando et al. (2017), and summarized in Section 3.1.

### 3.1 Dataset and Preprocessing

We use the image set BBBC021v1 (Caie et al., 2010) available from the Broad Bioimage Benchmark Collection (Ljosa et al., 2012). This dataset corresponds to cells prepared on 55 plates across 10 separate batches, and imaged in three color channels (i.e. stains); for a population of control cells, a compound (DMSO) with no anticipated drug effect was applied, while various other drug compounds were applied to the remaining cells. We compute the corresponding embeddings for each cell image using the method in Ando et al. (2017), summarized as follows. For a 128 by 128 pixel crop around each cell for each of the three color channels, a Deep Metric Network generates a 64-dimensional embedding vector. The three vectors corresponding to the three color channels are concatenated, forming a 192-dimensional embedding for each cell image. Using the embedding vectors for all cells, a Typical Variation Normalization (TVN) is applied in which the negative controls (i.e., DMSO) are whitened. Specifically, in the principal component analysis (PCA) basis of only negative control cells, an affine transformation is found so that the negative controls have mean zero and identity covariance matrix. The same transformation is then applied to the embeddings of all cells. Note that Ando et al. (2017) uses a different terminology, where TVN includes an additional transformation named CORAL, which will be presented and compared with in Section 3.4.

We use the same subset of treatments (concentration of a particular compound) evaluated in Ljosa et al. (2013) and Ando et al. (2017). This subset has 103 treatments from 38 compounds, each belonging to one of 12 known mechanism of action (MOA) groups. Sample cell images from the 12 MOA groups are shown in Figure 1. In Figure 6, we show a heatmap of the cosine similarity matrix between pairs of the selected treatments for the TVN embeddings. This figure shows how embeddings of the same compound, and embeddings of the compounds with the same MOA have a tendency to cluster closer to each other in terms of the cosine distance.

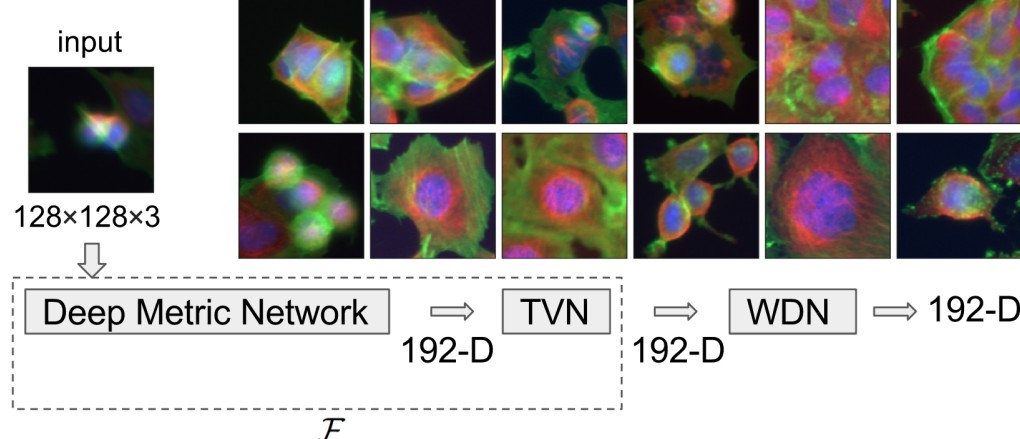

Figure 1: A flowchart describing the procedure we use to generate and remove nuisance variation from image embeddings. The embedding generation is described in Section 3.1 is characterized by $\mathcal{F}$, which maps each 128 by 128 color image into a 192-dimensional embedding vector. The nuisance variation removal by our method is denoted by WDN (Wasserstein Distance Network). The 12 images on the right side show representative images of cells treated with drug compounds with one of the 12 known mechanisms of action (MOA), from the BBBC021 dataset (Ljosa et al., 2012).

### 3.2 Evaluation Metrics

Our method is evaluated by three metrics, the first two of which measure how much biological signal is preserved in the transformed embeddings, and the last one of which measures how much nuisance variation has been removed.

### 3.2.1 k-Nearest Neighbor Mechanism of Action Assignment

Each compound in the BBBC021 dataset has a known MOA. A desirable property of embedding vectors is that compounds with the same MOA should group closely in the embedding space. This property can be assessed in the following way using the ground truth MOA labels for each treatment.

First, compute the mean $m_X$ of the embeddings for each treatment $X$ in each domain. Find the nearest $k$ neighbors $n_{X,1}, n_{X,2}, ..., n_{X,k}$ of $m_X$ either (i) not belonging to the same compound or (ii) not belonging to the same compound or batch (domain), and compute the portion of them having the same MOA as $m_X$. Our metric is defined as the average of this quantity across all treatment instances $X$ in all domains. If nuisance variation is corrected by transforming the embeddings, we may expect this metric to increase. The reason for excluding same-domain nearest neighbors is to avoid the in-domain correlations from interfering with the metric.

The nearest $k$ neighbors are found based on the cosine distance, which is more natural for the embedding space than the Euclidean distance, and can be directly compared with methods in existing literature. Moreover, our k-NN metrics are generalizations of the 1-NN metrics used in Ljosa et al. (2013) and Ando et al. (2017).

### 3.2.2 Silhouette Index

Cluster validation measures provide another way of characterizing how well compounds from the same MOA group together in embedding space. In our application, each 'cluster' is a chosen MOA containing a group of treatments, and each point in a cluster is the mean of embeddings for a particular treatment (i.e. compound and concentration) and domain.

The Silhouette index is one such measure that compares each point's distance from points in its own cluster to its distance from points in other clusters. It is defined as

$$s(i) = \frac{b(i) - a(i)}{\max\{a(i), b(i)\}}, \tag{12}$$

where $a(i)$ is the average distance from point $i$ to all other points in its cluster, and $b(i)$ is the minimum of all average distances from $i$ to all other clusters (i.e. the distance to the closest neighboring cluster) (Rousseeuw, 1987). The Silhouette index ranges between -1 and 1, with higher values indicating better clustering results.

### 3.2.3 Domain Classification Accuracy per Treatment

Another metric measures how well domain-specific nuisance information has been 'forgotten'. To do this, for each treatment we train a classifier to predict for each embedding the batch (domain) from the set of possible batches (domains) for that treatment. We evaluate both a linear classifier (i.e. logistic regression) and a random forest with 3-fold cross validation. If nuisance variation is being corrected, the batch (domain) classification accuracy should decrease significantly. Because only the negative control (i.e., DMSO) has replicates across experiment batches in our dataset, we train and evaluate these two batch classifiers on this compound only.

## 3.3 Procedure

### 3.3.1 Leave-One-Compound-Out Cross-Validation

For the model with either early stopping or a regularization term, the hyperparameters (i.e., the stopping time step or the regularization weights) can be selected by a cross-validation procedure to avoid overfitting (see Godinez et al. (2017) for an example). In particular, we apply this procedure to the case of early stopping. Each time, an individual compound is held out, and the stopping time step is determined by maximizing the average k-NN MOA assignment metric for $k = 1, ..., 4$ on the remaining compounds. Figure 2 illustrates the k-NN MOA assignment metrics as a function of time steps in the case when early stopping is used with compound mitoxantrone held out.

For the embeddings transformed at the optimal time step, we evaluate the k-NN MOA assignment metrics for the held-out compound. The procedure is repeated for all the compounds, and the k-NN MOA assignment metrics are aggregated across all the compounds. Intuitively, for each fold of

this leave-one-compound-out cross-validation procedure, the held-out compound can be treated as a new compound with unknown MOA, and the hyperparameters are optimized over the compounds with known MOAs. In our case, we find that the optimal time step remains the same, i.e., 28000, regardless of the held-out compound.

### 3.3.2 STANDARD ERRORS OF THE METRICS

To assess whether the improvements in the k-NN MOA assignment metric and the Silhouette index are statistically significant, we estimate the standard errors of the metrics using a nonparametric bootstrap method. Each time, the bootstrap samples are generated by sampling with replacement the embeddings preprocessed by TVN in each well, and the metrics are evaluated using the bootstrap samples. We repeat the procedure for 200 times, and obtain the standard errors of the 200 bootstrap estimates of the metrics, which are summarized in Tables 1 and 3.

### 3.3.3 TRAINING PROCEDURE

The embedding transformations $A_d(x) = M_d x + b_d$ are initialized to $M_d = I$, $b_d = 0$, since we wish for the learned transformations to be not too far from the identity transformation.

To approximate each of the Wasserstein functions $f_{t,d_i,d_j}$ in (eq. 7), we use a network consisting of softplus layer followed by a scalar-valued affine transformation. The softplus loss is chosen because the Wasserstein distance estimates it produces are less noisy than other kinds of losses and it avoids the issue of all neurons becoming deactivated (which can occur for example when using RELU activations).

The dimension of the softplus layer used to approximate each Wasserstein function is 2. Optimization is done using stochastic gradient instead of the sums in (eq. 7). For simplicity, the minibatch size for each treatment per iteration step is fixed throughout. In the results presented, the minibatch size is $50$. Optimization for both classes of parameters $\theta_T$ and $\theta_W$ is done using separate RMSProp optimizers. Prior to training $\theta_T$, we use a 'pre-training' period of 20000 time steps to obtain a good approximation for the Wasserstein distances. After this, we alternate between adjusting $\theta_T$ for $40$ time steps and optimizing over $\theta_W$ for a single time step.

## 3.4 RESULTS

We compare our results to either using no transformation other than normalization (TVN) and CORAL. CORAL applies a domain-specific affine transformation to the embeddings represented as the rows of a matrix $X_d$ from domain $d$ in the following way. On the negative controls only, the covariance matrix across the entire experiment $C$ as well as the covariance $C_d$ in each domain $d$ are computed. Notice that since TVN had already been applied (see Section 3.1), $C = I$. Then, all embedding coordinates in domain $d$ are aligned by matching the covariance structures. Alignment is done by computing the new embeddings $X_d^{\text{aligned}} = X_d R_d^{-1/2} R^{1/2}$. Here $R_d = C_d + \eta I$ and $R = C + \eta I$ are regularized covariance matrices, with the regularizer $\eta = 1$, which is the same as that in Ando et al. (2017).

Other variations of the training procedure are discussed in Sections 3.4.3 and 3.5.

### 3.4.1 VISUALIZATION OF RESULTS

Figure 3 shows the first two principal components of the embeddings transformed by WDN, compared with the embeddings preprocessed by TVN (see Section 3.1) and the embeddings generated by the CORAL method proposed in Sun et al. (2017) and applied by Ando et al. (2017).

Figure 5 shows the dosage response for each compound based on each set of transformed embeddings. WDN is seen to better preserve the geometry of the embeddings than CORAL.

### 3.4.2 METRICS

Table 1 shows the k-NN MOA assignment metrics of our transformed embeddings (early stopping and some particular choices of the regularization weights) compared to the original embeddings as

well as the estimated standard errors. We also include the values of this metric for CORAL. We find that our WDN method performs better than CORAL in terms of the k-NN MOA assignment metrics.

Finally, Table 2 compares the average batch classification accuracy for a linear classifier (i.e., logistic regression) and a random forest classifier for the original TVN embeddings, WDN embeddings (early stopping and some particular choices of the regularization weights), CORAL embeddings, and for reference, a trivial transformation for which all embeddings are set to zero. For each run, given a classifier and a transformed set of embeddings, we compute the mean accuracy for that classifier using 3-fold cross validation. We see that the batch classification accuracy for the embeddings using our method is substantially smaller than that using TVN or CORAL, indicating our method is removing nuisance variation.

### 3.4.3 EARLY STOPPING VERSUS REGULARIZATION TERM

We have tried regularizing the network either with a regularization term or early stopping. When using a regularization term, the loss function and the evaluation metrics converge for a chosen set of regularization weights. We present the resulting k-NN MOA assignment metrics in Table 1 for several values of $\lambda = \lambda_M = \lambda_b$, as well as for the early stopping at the optimal time step 28000. We see that the smaller regularization ($\lambda = 40$) results in a greater removal of nuisance variation. However, removing more nuisance variation may be counterbalanced by also removing relevant biological signal, as suggested by the k-NN MOA assignment metrics in Table 1. In addition, using a non-optimal choice of the regularization weight may result in a lower Silhouette index, as shown in Table 3.

Each approach has advantages and disadvantages. Using early stopping is simpler and does not require a computationally intensive grid search over all parameters to obtain optimal results, but on the other hand this may be a limiting factor in performance because of the smaller selection of parameters. If the transformed embedding vectors do not follow an approximately direct path throughout the optimization, early stopping may miss the optimal solution. This development is likely not a problem in our applications, since the transformation is small. This explains why early stopping does not seem to produce negative side effects. We find that early stopping produces a better result in terms of the k-NN MOA assignment metrics than the values of $\lambda$ we have tried, but we anticipate using a more thorough search over the regularization weights would yield similar results between the two methods.

The learning curves for both the early stopping case and some regularization weights are shown in Figure 4.

### 3.5 ADDITIONAL EXPERIMENTS

To assess how the hyperparameters of the model affect its performance, we conduct additional experiments by varying the hyperparameters. For example, the minibatch size is increased from 50 to 100. The results are similar except that the learning curve in the case of 100 appears less noisy. Moreover, the architecture of the network that estimates the pairwise Wasserstein distances is made more complicated by increasing the number of hidden layers from two to three and four, and the number of nodes per layer from two to four and eight, respectively. Again, there is no significant difference in the results except that the curves of the k-NN MOA assignment metrics over the number of time steps appeared more stable.

## 4 CONCLUSION

We have shown how a neural network can be used to transform embedding vectors to 'forget' specifically chosen domain information as indicated by our proposed domain classification metric. The transformed embeddings still preserve the underlying geometry of the space and improve the k-NN MOA metrics. Our approach uses the Wasserstein distance and can in principle handle fairly general distributions of embeddings (as long as the neural network used to approximate the Wasserstein function is general enough). Importantly, we do not have to assume that the distributions are Gaussian.

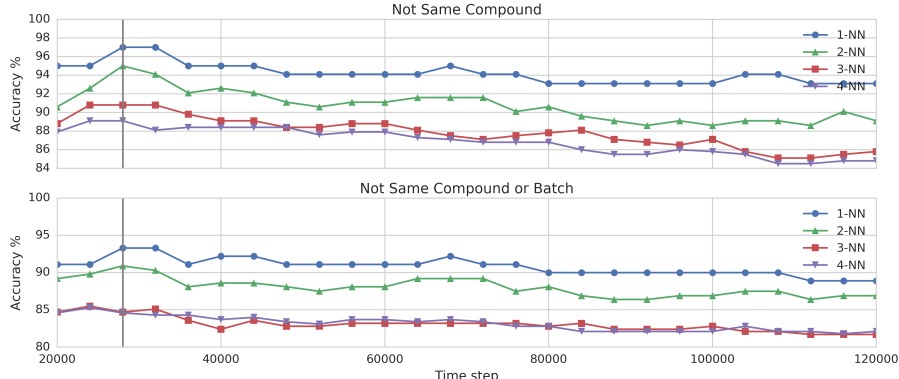

Figure 2: k-NN MOA assignment metrics for one fold of the cross-validation with compound mitoxantrone held out. The average k-NN metric is used to select the stopping time step, which is at 28000 as indicated by the vertical line. Early stopping is supposed to preserve relevant biological signal while remove batch-level nuisance variation. The top panel shows the 'not same compound' metric, and the bottom panel shows the 'not same compound or batch' metric (see Section 3.2.1 for details).

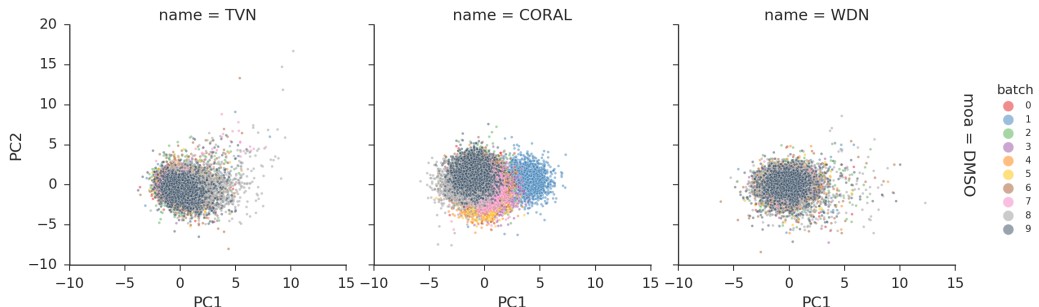

Figure 3: Comparison of the first two principal components for the embeddings of the negative control (i.e., DMSO) after preprocessing (i.e., TVN) (left), embeddings transformed by the TVN + CORAL method proposed in Ando et al. (2017) (middle), and embeddings transformed by TVN + WDN (right), which illustrates the reduction of batch-level nuisance variation. Each color corresponds to a batch, and there are ten batches in total. Our method is designed to match embeddings of compounds across batches while not distorting the geometry of the embedding space.

The framework itself is quite general and extendible (see Section 4.1). Unlike methods that use only the controls for adjusting the embeddings, our method can also utilize information from replicates of a treatment across different domains. However, the dataset used did not have treatment replicates across batches, so we only relied on aligning based on the controls. Thus we implicitly assume that the transformation for the controls matches that of the various compounds. We expect our method to be more useful in the context of experiments where many replicates are present, so that they can all be aligned simultaneously. We expect transformations learned for such experiments to have better generalizability since it would be using available knowledge from a greater portion of the embedding space.

Our approach requires a choice of free parameters, either for regularization or early stopping, which we address by cross validation across compounds. We discuss potential future directions below, as well as other limiting issues.

Table 1: k-NN MOA assignment metrics for TVN only, TVN + WDN, and TVN + CORAL, where WDN is regularized by either early stopping or a regularization term with weights. The last column shows the standard errors estimated by the bootstrap method described in Section 3.3.2. These metrics suggest that WDN with early stopping yields embeddings containing stronger biological signal than TVN or TVN + CORAL. Here the $\lambda = 40, 80, 160$ are the WDN transformation after converging, when a regularization term $\lambda = \lambda_M = \lambda_b$ is added.

| k-NN | TVN only | WDN | CORAL | $\lambda = 40$ | $\lambda = 80$ | $\lambda = 160$ | $\pm$ |
|---|---|---|---|---|---|---|---|
| 1 | 96.1% | 97.1% | 97.1% | 97.1% | 97.1% | 96.1% | 1.1% |
| 2 | 92.2% | 95.1% | 94.7% | 93.7% | 95.1% | 93.7% | 0.7% |
| 3 | 89.3% | 91.9% | 90.9% | 90.6% | 91.9% | 91.6% | 0.6% |
| 4 | 87.4% | 89.9% | 88.4% | 88.9% | 89.2% | 89.7% | 0.5% |

(a) Not same compound

| k-NN | TVN only | WDN | CORAL | $\lambda = 40$ | $\lambda = 80$ | $\lambda = 160$ | $\pm$ |
|---|---|---|---|---|---|---|---|
| 1 | 90.2% | 93.5% | 91.3% | 92.4% | 91.3% | 91.3% | 1.0% |
| 2 | 88.9% | 91.1% | 89.4% | 90.6% | 90.6% | 89.4% | 0.9% |
| 3 | 84.7% | 86.2% | 85.1% | 84.0% | 85.1% | 85.4% | 0.7% |
| 4 | 83.9% | 84.8% | 84.5% | 83.6% | 84.5% | 84.5% | 0.6% |

(b) Not same compound or batch

Table 2: Batch (domain) classification accuracy after transformations for controls using either logistic regression (LR) or a random forest (RF) with 3 folds. We only use controls here because other treatments have no replicates across batches in our dataset. We compare TVN only, TVN + WDN (with early stopping), and TVN + CORAL. We also show the results for WDN with $\lambda = 40, 80, 160$ for $\lambda = \lambda_M = \lambda_b$. The 'trivial transformation' (send all embeddings to a point) is provided for reference. If all nuisance information is removed, the batch accuracy would drop that of the trivial transformation. The table below shows that WDN removes some of the nuisance variation, at least from the controls.

| | TVN only | WDN | CORAL | $\lambda = 40$ | $\lambda = 80$ | $\lambda = 160$ | Trivial trans. |
|---|---|---|---|---|---|---|---|
| LR | $63.6 \pm 1\%$ | $39.8 \pm 0.6\%$ | $66.4 \pm 0.7\%$ | $28.0 \pm 0.8\%$ | $46.8 \pm 0.9\%$ | $56.2 \pm 0.9\%$ | 16.6% |
| RF | $45.9 \pm 0.2\%$ | $34.4 \pm 0.7\%$ | $46.8 \pm 0.6\%$ | $26.7 \pm 0.7\%$ | $33.3 \pm 0.7\%$ | $39.5 \pm 0.1\%$ | 16.6% |

Table 3: We show the silhouette index for TVN only, TVN + WDN, and TVN + CORAL, as discussed in Section 3.2.2. Here WDN refers to the the result using early stopping, and $\lambda = 40, 80, 160$ refers to the result when using a regularization with $\lambda = \lambda_M = \lambda_b$. Both WDN and CORAL appear to increase the cohesion, as measured by this index. The estimated error denoted by $\pm$ was determined by the bootstrapping procedure described in Section 3.3.2

| | TVN only | WDN | CORAL | $\lambda = 40$ | $\lambda = 80$ | $\lambda = 160$ | $\pm$ |
|---|---|---|---|---|---|---|---|
| Silhouette index | 0.5042 | 0.5126 | 0.5099 | 0.5088 | 0.5115 | 0.5093 | 0.0019 |

## 4.1 FUTURE WORK

One possible modification we considered would be to replace the form of the cost function by the following, which would more closely resemble finding the Wasserstein barycenter:

$$\sum_{i,j=1}^{N} W(\nu_i, A_{d_j}(\nu_j)). \tag{13}$$

The difference is that instead of comparing the pairwise transformed distributions, we instead compare the transformed distributions to the original distributions. One advantage for this approach is that it avoids the 'shrinking to a point' problem, and therefore does not require a regularization term or early stopping to converge to a meaningful solution. However, we did not find better performance for the new form of the cost function (eq. 13) for our specific dataset.

An alternative regularization term to the one we used penalizing how much the transformation differs from the identity may be used. One interesting choice might be to penalize the change of pairwise distances between treatments within a specific domain. Intuitively, in-domain variations carry biological signal that we would like to preserve, and using such a regularization term does so explicitly.

The Wasserstein functions were approximated with very simple nonlinear functions, and it is possible better results would be obtained using more sophisticated functions capturing the Wasserstein distance and its gradients more accurately. Similarly, The transformations $A_d$ could be generalized from affine to a more general class of functions. As in Shaham et al. (2017), we expect residual networks would make natural candidates for these transformations.

One possibility is to fine-tune the Deep Metric Network used to generate the embeddings instead of training a separate network on its outputs (or perhaps several such networks for the separate image stains used).

Another issue is how to weigh the various Wasserstein distances against each other. This might improve the results if there are many more points from some distributions than others (which happens in the real data). Further, it is unclear how a regularization term should be weighed against the Wasserstein loss terms.

Another extension may involve applying our method hierarchically on the various domains of the experiment. However, this would require replicates on multiple hierarchical levels.

Since the k-NN MOA assignment metric is based on the cosine distance, it is possible better results could be obtained by modifying the metric used to compute the Wasserstein distance accordingly, e.g. finding an optimal transportation plan only in non-radial directions.

ACKNOWLEDGMENTS

We would like to thank Mike Ando, Marc Coram, Marc Berndl, Subhashini Venugopalan, Arunachalam Narayanaswamy, Yaroslav Ganin, Luke Metz, Eric Christiansen, Philip Nelson, and Patrick Riley for useful discussions and suggestions.

## A    LEARNING CURVES WITH AND WITHOUT REGULARIZATION

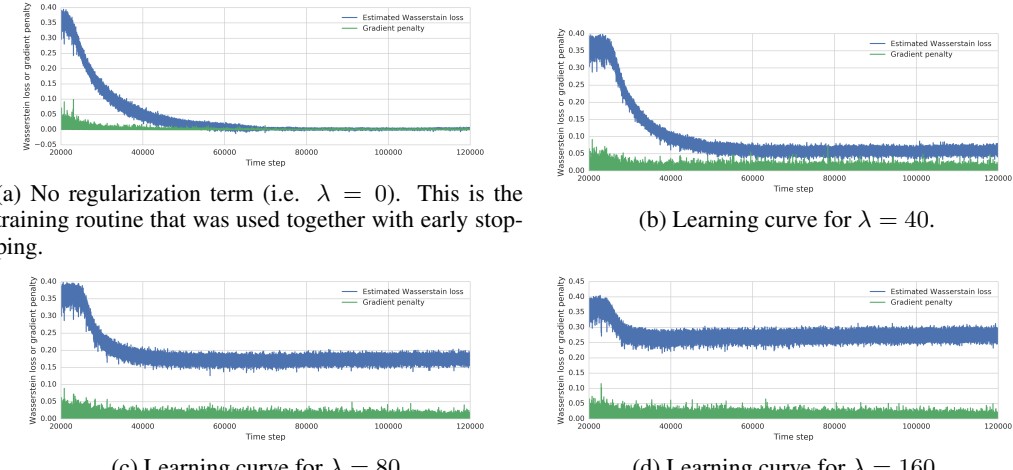

(a) No regularization term (i.e. $\lambda = 0$). This is the training routine that was used together with early stopping.

(b) Learning curve for $\lambda = 40$.

(c) Learning curve for $\lambda = 80$.

(d) Learning curve for $\lambda = 160$.

Figure 4: Sample learning curves for WDN with regularization term for $\lambda = \lambda_M = \lambda_b = 0, 40, 80, 160$. The Wasserstein loss and the gradient penalty term as a function of the number of time steps trained on BBBC021 image dataset, after the Wasserstein parameters have been pretrained for 20000 steps. The larger the regularization weight, the further the point of convergence is from zero.

## B  DOSAGE RESPONSE PLOTS

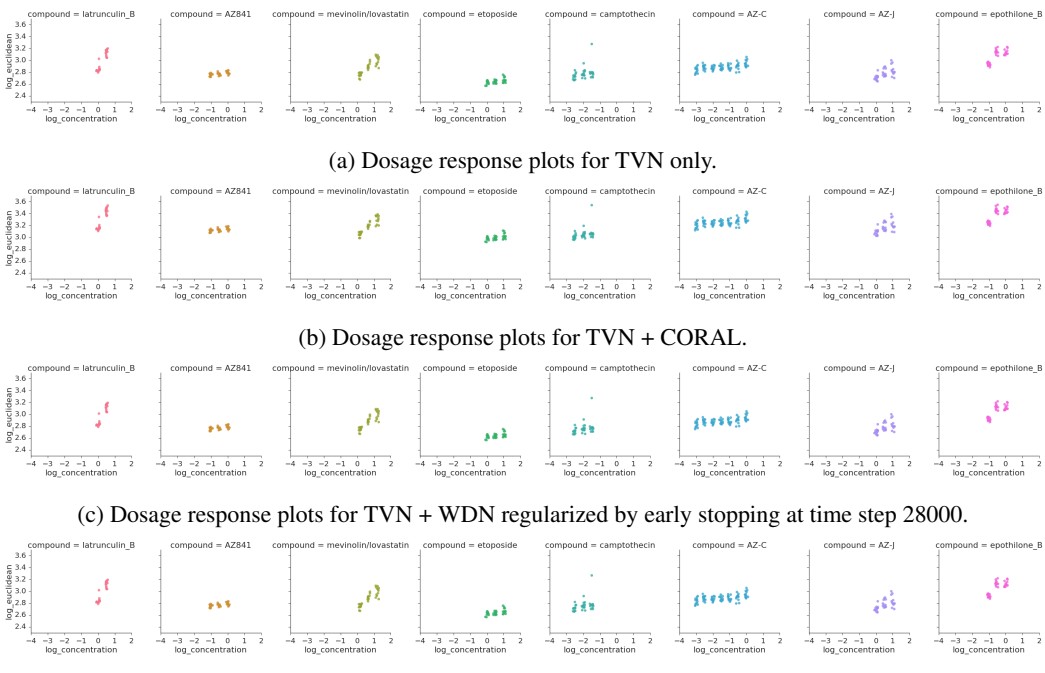

(a) Dosage response plots for TVN only.

(b) Dosage response plots for TVN + CORAL.

(c) Dosage response plots for TVN + WDN regularized by early stopping at time step 28000.

(d) Dosage response plots for TVN + WDN with regularization term for $\lambda = \lambda_M = \lambda_b = 80$.

Figure 5: Dosage response curves for each compound, as evaluated by the natural logarithm of the Euclidean distance of the embeddings from the origin (i.e. the center of the negative control). These plots show that WDN better preserves the geometry of the embedding space than CORAL, where the latter can magnify the scale of the response. WDN regularized by early stopping and a regularization term both slightly alter the embeddings.

# C  COMPOUND SIMILARITY MATRIX

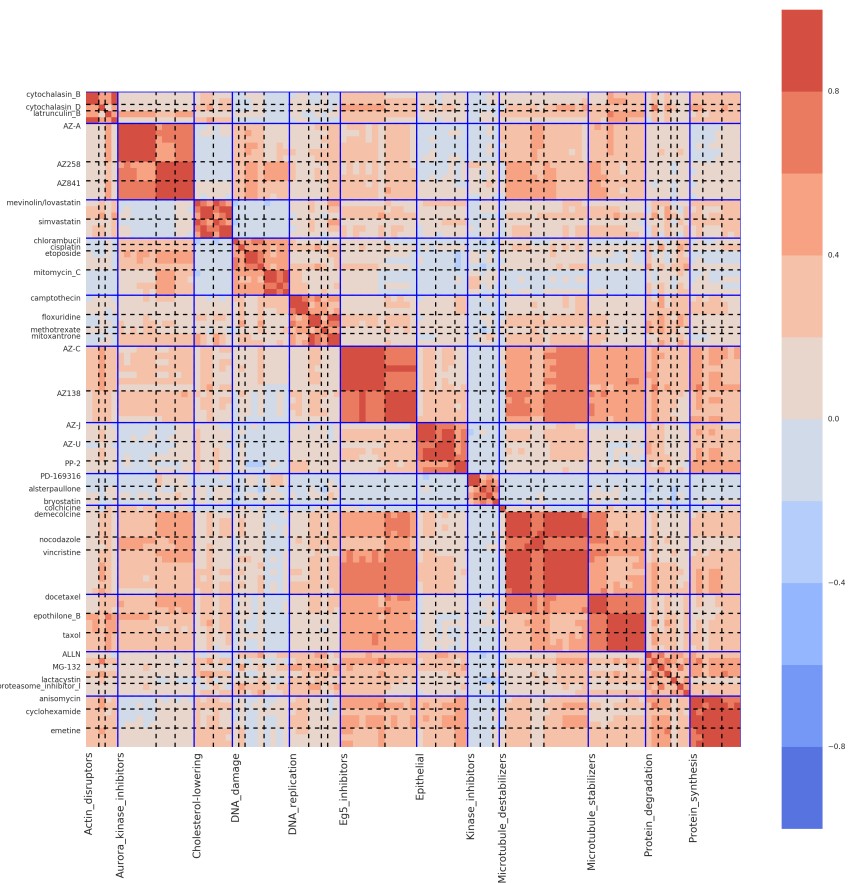

Figure 6: A heatmap showing the cosine similarity matrix between pairs of treatments for the TVN embeddings. Same-MOA compounds are grouped together, and the blue lines show distinctions between different MOAs. The block diagonal terms correspond to the similarity matrices for same-MOA compounds. This plot shows how same-MOA compounds tend to be more closely clustered together in the embedding space.

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
