# OpenReview forum: "Correcting Nuisance Variation using Wasserstein Distance"
_ICLR.cc/2018/Conference — Reject_

### Official Review · AnonReviewer1 · 2017-11-27
**The paper discusses a method for adjusting image embeddings in order tease apart technical variation from biological signal.**

**Rating:** 5
**Confidence:** 3

**Review:**

The paper discusses a method for adjusting image embeddings in order tease apart technical variation from biological signal. A loss function based on the Wasserstein distance is used.
The paper is interesting but could certainly do with more explanations.

Comments:
1. It is difficult for the reader to understand a) why Wasserstein is used and b) how exactly the nuisance variation is reduced.
A dedicated section on motivation is missing.

2. Does the Deep Metric network always return a '64-dim' vector?
Have you checked your model using different length vectors?

3. Label the y-axis in Fig 2.

4. The fact that you have early-stopping as opposed to a principled regularizer also requires further substantiation.

---

> ### Author Response · Authors · 2018-01-05
> **Response to Reviewer**
>
> We appreciate the comments and suggestions of all the reviewers, which we agreed pointed out important ways in which our work could be improved. We have amended our manuscript with more thorough descriptions and explanations, and added new results. Specific responses to the reviewer is given below:
>
> 1. It is difficult for the reader to understand a) why Wasserstein is used and b) how exactly the nuisance variation is reduced. A dedicated section on motivation is missing.
>
> We have expanded a section to include a qualitative description of the Wasserstein distance and a motivating concept for our work, the Wasserstein barycenter. The idea of Wasserstein barycenter is two-fold. First, we want to match the distributions after transformation. Second, we want the perturbation as small as possible. These two ideas are reflected in our method. Although other metrics may also be used, the Wasserstein distance and similar related metrics capture relevant geometric information between the distributions. We have extended our description of the Wasserstein distance and its application in our paper.
>
> 2. Does the Deep Metric network always return a '64-dim' vector? Have you checked your model using different length vectors?
>
> The deep neural network we used always returns a 64-dimensional vector by design, although we do not expect the results to be sensitive to the dimensionality, as long as there are sufficiently many data points compared to the inherent dimension of the data. Additionally, we have experimented with our network on synthetic low-dimensional data as well, obtaining the expected results.
>
> 3. Label the y-axis in Fig 2.
>
> Done.
>
> 4. The fact that you have early-stopping as opposed to a principled regularizer also requires further substantiation.
>
> We initially used early stopping mostly to show our framework works as a proof-of-concept. Early stopping provides a simpler framework under which fewer parameters required to be optimized (i.e. the early stopping time instead of the separate regularization weights used to limit the transformation). In our revised manuscript we have included experiments carried out with a regularizer. Our results showed that the results from early stopping were comparable, and even better than the experiments we carried out with a regularizer, which did not have their hyperparameters fine-tuned. We agree that a principled regularizer may yield better results once all of its hyperparameters have been optimized.  However, tuning the hyperparameters is a potentially difficult problem outside the scope of this work.

---

### Official Review · AnonReviewer2 · 2017-11-27
**Wasserstein distances for eliminating batch effect, not enough novelty and no thorough comparisons to other methods.**

**Rating:** 4
**Confidence:** 5

**Review:**

The authors present a method that aims to remove domain-specific information while preserving the relevant biological information between biological data measured in different experiments or "batches". A network is trained to learn the transformations that minimize the Wasserstein distance between distributions. The wasserstein distance is also called the "earth mover distance" and is traditionally formulated as the cost it takes for an optimal transport plan to move one distribution to another. In this paper they have a neural network compute the wasserstein distance using a different formulation that was used in Arjovsky et al. 2017, finds a lipschitz function f, which shows the maximal difference when evaluated on samples from the two distributions. Here these functions are formulated as affine transforms of the data with parameters theta that are computed by a neural network. Results are examined mainly by looking at the first two PCA components of the data.


The paper presents an interesting idea and is fairly well written. However I have a few concerns:
1. Most of the ideas presented in the paper rely on works by Arjovsky et al. (2017), Gulrajani et al. (2017), and Gulrajani et al. (2017). Some selections, which are presented in the papers are not explained, for example, the gradient penalty, the choice of \lambda and the choice of points for gradient computation.
2. The experimental results are not fully convincing, they simply compare the first two PC components on this Broad Bioimage benchmark collection. This section could be improved by demonstrating the approach on more datasets.
3. There is a lack comparison to other methods such as Shaham et al. (2017). Why is using earth mover distance better than MMD based distance? They only compare it to a method named CORAL and to Typical
Variation Normalization (TVN). What about comparison to other batch normalization methods in biology such as SEURAT?
4. Why is the affine transform assumption valid in biology? There can definitely be non-linear effects that are different between batches, such as ion detection efficiency differences.
5. Only early stopping seems to constrain their model to be near identity. Doesn't this also prevent optimal results ? How does this compare to the near-identity constraints in resnets in Shaham et al. ?

---

> ### Author Response · Authors · 2018-01-05
> **Response to Reviewer (comments 1-2)**
>
> We appreciate the comments and suggestions of all the reviewers, which we agreed pointed out important ways in which our work could be improved. We have amended our manuscript with more thorough descriptions and explanations, and added new results. Specific responses to the reviewer is given below:
>
> 1. Most of the ideas presented in the paper rely on works by Arjovsky et al. (2017), Gulrajani et al. (2017), and Gulrajani et al. (2017). Some selections, which are presented in the papers are not explained, for example, the gradient penalty, the choice of \lambda (now replaced by \gamma) and the choice of points for gradient computation.
>
> We would like to point out that while we do rely on the methods of Arjovsky et al. (2017), Gulrajani et al. (2017) for estimating the Wasserstein distance, the application of these methods for correcting nuisance variation is novel and independent of these methods. This aspect of our work is a novel way of removing nuisance variation, a significant problem in high-throughput biological experiments. Our approach is based on minimizing the sum of pairwise Wasserstein distances of a transformed set of coordinates. This method is inspired by, but distinct from, finding the Wasserstein barycenter. We hope that our approach demonstrates a novel and general framework for removing nuisance variation.
>
> We have added more thorough explanations of our approach for approximating the Wasserstein distance, including the choice of \lambda and the choice of points for the gradient computation.
>
> 2. The experimental results are not fully convincing, they simply compare the first two PC components on this Broad Bioimage benchmark collection. This section could be improved by demonstrating the approach on more datasets.
>
> While we use the first two PC components to illustrate the effect of our transformation, we rely on other quantitative metrics for evaluating the performance of our framework. Specifically, we use domain classification accuracy to assess the extent to which nuisance variation has been removed (discussed in section 3.2.3 and shown in table 2). We also included the k-NN MOA assignment metric to evaluate the quality of the transformed embeddings (discussed in section 3.2.1 and shown in table 1).
>
> In addition, in the revised manuscript we added another quantitative metric, the average silhouette index of the MOAs to better evaluate the effects of our transformation (see section 3.2.2 and table 3).
>
> In our original manuscript the k-NN MOA metric did not show significant differences among the evaluated methods when using cross validation leaving out half the compounds at a time. This occurred because there were not enough compounds remaining in each test/evaluation cross validation folds. However, in our revised manuscript our new analysis using a leave-one-compound-out cross validation showed that the framework can be used to attain a significant improvement in the k-NN MOA metric. Using leave-one-compound-out cross validation has been used in other studies [Godinez, William J., et al. “A Multi-Scale Convolutional Neural Network for Phenotyping High-Content Cellular Images.” Bioinformatics (2017)]. This method represents a more realistic setting when a compound with unknown MOA takes the role of the held-out compound.
>
> The reasons we base our results on specifically the BBBC021 dataset in this paper are:
>
> This is an open dataset that has been used as a standard. We would like to produce a direct comparison with existing literature. These include:
>
> 1. Ljosa, Vebjorn, et al. "Comparison of methods for image-based profiling of cellular morphological responses to small-molecule treatment." Journal of biomolecular screening 18.10 (2013): 1321-1329.
> 2. Ando, D. Michael, Cory McLean, and Marc Berndl. "Improving Phenotypic Measurements in High-Content Imaging Screens." bioRxiv (2017): 161422.
> 3. Pawlowski, Nick, et al. "Automating Morphological Profiling with Generic Deep Convolutional Networks." bioRxiv (2016): 085118.
> 4. Singh, S., et al. "Pipeline for illumination correction of images for high‐throughput microscopy." Journal of microscopy 256.3 (2014): 231-236.
>
> We have tested our procedure on another dataset with promising results, but unfortunately we are unable to release it at this time.
>
> In addition, we have added the dosage response plots before and after the transformation. These plots show qualitatively that our transformation preserves the structural dosage-response.
>
> Points 3-5 will be addressed in a separate comment due to the character limit.

---

> ### Author Response · Authors · 2018-01-05
> **Additional Comments in Response (comments 3-4)**
>
> 3. There is a lack comparison to other methods such as Shaham et al. (2017). Why is using earth mover distance better than MMD based distance? They only compare it to a method named CORAL and to Typical Variation Normalization (TVN). What about comparison to other batch normalization methods in biology such as SEURAT?
>
> We added a more thorough motivation in our manuscript for motivating the usage of the earth mover distance.
>
> The MMD distance is closely related to the earth mover distance. Specifically, the earth mover distance between two distributions \nu_r and \nu_g is equal to
>
> \sup_{\|f\|_L \le1} E_{x\sim \nu_r} f(x) - E_{x\sim \nu_g} f(x).
>
> Above f belongs to the space of Lipschitz functions with constant 1 (i.e. f is a contraction).
>
> To compute the MMD, we suppose first that we have a kernel function k: \chi \times \chi \to mathbb{R} with an associated reproducing kernel Hilbert space \mathcal{H}, and the condition that f is Lipschitz-1 is replaced by the condition that the \mathcal{H}-1 norm of f is bounded by 1. Under this condition, Shaham et al. (2017) provide a sample estimate for the MMD distance, based on the supposed kernel. In their work, the kernel is constructed as the sum of three Gaussian kernels with variances \sigma_i chosen to be m/2 , m, and 2m, where m is the median of the average distance between a point in the target sample to its nearest 25 neighbors.
>
> In our framework, we do not have a `source' and `target’ distribution: instead, we match an arbitrary number of distributions indexed by both treatment and domain. Therefore, to apply the MMD sample estimate of Shaham et al. (2017), we would have to continuously update the chosen variances \sigma_i. We think that applying this sample estimate in our framework may be fruitful and give similar results to our method, but because of the added complexity of fitting this cost function into our framework we believe this experiment is outside the scope of this paper.
>
> The alignment method in SEURAT we believe the reviewer is referring to is described here:
> https://www.biorxiv.org/content/biorxiv/early/2017/07/18/164889.full.pdf
>
> The SEURAT method we think the reviewer is referring to is based on applying canonical-correlation analysis between two datasets, followed by `dynamic time warping’ to correct for changes in density. As for the application in Shaham et al. (2017), the method is specific to aligning two datasets. In our paper, we are interested in the case of many domains, so it is not clear to us how to directly compare our results with those in SEURAT.
>
> 4. Why is the affine transform assumption valid in biology? There can definitely be non-linear effects that are different between batches, such as ion detection efficiency differences.
>
> For the dataset of interest, we applied a universal transformation to the embeddings such that embeddings for negative controls are standardized to have 0 mean and unit variance in all coordinates. We observed that increasing dosages of each compound had embeddings shifted in roughly the same direction by increasing amounts, and the variances along the largest principal axes also increased in a manner consistent with the embeddings undergoing an affine transform. In this setting, we elected to model the impact of batch effects by affine transforms, the intuition being that we can think of batch effects as resembling small, random, drug-like perturbations resulting from unobserved covariates. This result can be motivated if we assume that the perturbations of applying a treatment generally have small effects for the embeddings. We do not expect this assumption to hold generally. The manuscript has been updated to reflect this motivation.
>
> We hope in the future to test relaxing this assumption, or instead even to fine-tune the original deep neural network to control nuisance variation.

---

> ### Author Response · Authors · 2018-01-05
> **Additional Comments in Response (comment 5)**
>
> 5. Only early stopping seems to constrain their model to be near identity. Doesn't this also prevent optimal results ? How does this compare to the near-identity constraints in resnets in Shaham et al. ?
>
> We used early stopping to show our framework works as a proof-of-concept, and we agree it is not guaranteed to yield optimal results. We agree that a principled regularizer may in principle yield better results, because (i) the larger space of penalty hyperparameters and (ii) the potentially non-direct optimization path in the case of a penalty term. In our revised manuscript we have included experiments carried out with a regularizer, where the transformation was constrained to be close to the identity, similarly to Shaham et al. We have found that early stopping was comparable to a penalty term (actually performed better). However, we have not done an extensive search in the space of penalty hyperparameters. We expect that even once the penalty hyperparameters will be fine-tuned, the results will be comparable between the early stopping and regularizer cases, but not much better. This is because the perturbation was quite small, and therefore we expect that as learning progresses the transformed embeddings move in approximately straight paths. We have added the above comments to our manuscript (section 3.4.3).

---

### Official Review · AnonReviewer3 · 2017-11-28
**Review of Correcting Nuisance Variation using Wasserstein Distance**

**Rating:** 7
**Confidence:** 3

**Review:**

This contribution deal with nuisance factors afflicting biological cell images with a domain adaptation approach: the embedding vectors generated from cell images show spurious correlation. The authors define a Wasserstein Distance Network to find  a suitable affine transformation that reduces the nuisance factor. The evaluation on a real dataset yields correct results, this approach is quite general and could be applied to different problems.

The contribution of this approach could be better highlighted. The early stopping criteria tend to favor suboptimal solution, indeed relying on the Cramer distance is possible improvement.

As a side note, the k-NN MOA is central to for the evaluation of the proposed approach. A possible improvement is to try other means for the embedding instead of the Euclidean one.

---

> ### Author Response · Authors · 2018-01-05
> **Response to Reviewer**
>
> We appreciate the comments and suggestions of all the reviewers, which we agreed pointed out important ways in which our work could be improved. We have amended our manuscript with more thorough descriptions and explanations, and added new results. Specific responses to the reviewer is given below:
>
> 1. The contribution of this approach could be better highlighted. The early stopping criteria tend to favor suboptimal solution, indeed relying on the Cramer distance is possible improvement.
>
> We remark that our main goal was to introduce a general flexible framework, and using the Wasserstein was a demonstration of a specific choice that can yield substantial improvement. In our approach, we do not separate a `target’ and `source’ distribution, may include many domains, and can have different treatments across the various domains. We highlight that other distances may be used in our framework, such as the Cramer distance or the MMD distance. This may be preferable since the Cramer distance has unbiased sample gradients (Bellemare et al. 2017). Using the Cramer distance could reduce the number of steps required to adjust the Wasserstein distance approximation for each step of training the embedding transformation.
>
> We have added a discussion about early stopping versus a penalty term in our manuscript (section 3.4.3).
>
> To address the issue of potentially overfitting the stopping time, we have included a cross validation procedure based on holding out a single compound at a time. We found that the optimal stopping time was consistent regardless of the choice of the held-out compound. We discuss this in more detail in section 3.3.1.
>
> 2. As a side note, the k-NN MOA is central to for the evaluation of the proposed approach. A possible improvement is to try other means for the embedding instead of the Euclidean one.
>
> We remark that one of the main reason we used the k-NN MOA metric is to compare our work with previous approaches in existing literature, and the reviewer is correct to point out there may be better ways to improve this metric both for validation of the quality of embeddings as well as making MOA predictions for unknown compounds. In our dataset, we also expect that the embeddings for each treatment are sufficiently localized (as vectors, they are close to each other in the sense of having similar length and angle), so that the choice of centroid type would not alter them much. In the case when the embeddings are not sufficiently localized, one alternative would be to use an estimator for the Frechet mean [Salehian, Hesamoddin, et al. "An efficient recursive estimator of the Fréchet mean on a hypersphere with applications to medical image analysis." Mathematical Foundations of Computational Anatomy (2015)].

---

### Author Response · Authors · 2018-01-05
**Summary of Changes to Manuscript**

We summarize the main updates we made in our manuscript. For more details please see also the our responses to the reviewers.
1. In Section 2.2 we expanded our description and motivation for our framework.
2. In Section 2.3.1 we added motivation for our choice of transformation.
3. We revised Section 2.3.2 to include more details about how a regularizer may be used in our framework. In the same section we also included more detailed explanations, including for the gradient penalty, the choice of lambda, and the choice of points for gradient computation.
4. In Section 3.2.1 we clarified how the k-NN metric was computed.
5. We added a new metric (the silhouette metric), described in Section 3.2.2.
6. We modified our cross-validation procedure. Instead of using two folds over the compounds, we now use a leave-one-compound-out procedure. This is described in Section 3.3.1. The new method results in more sensitively detecting improvements for our metrics.
7. We estimated the standard error in some of the metrics using a bootstrapping procedure, described in Section 3.3.2.
8. In addition to early stopping, we tried using a penalty term for several hyperparameters. The resulting differences are discussed in Section 3.4.3.
9. We tried additional experiments which we describe in Section 3.5.
10. We added new potential improvements and modifications in Section 4.1.
11. We added the learning curves for various values of penalty hyperparameters in Figure 4 in Appendix A.
12. We added plots showing the dosage response for different transformation in Figure 5 in Appendix B.
13. We added a heatmap of the similarity matrix of the embedding space in Figure 6 in Appendix C.

---

### Decision · Program_Chairs · 2018-01-29
**ICLR 2018 Conference Acceptance Decision**

**Decision:**

Reject

**Comment:**

This is a nice but very narrow study of domain invariance in a microscopic imaging application.  Since the problem is very general, the paper should include much more substantial context, e.g. discussion of various alternative methods (e.g. the ones cited in Sun et al. 2017).  In order to contribute to the broader ICLR community, ideally the paper would also include application to more than just the one task.